# The Gut of Healthy Infants in the Community as a Reservoir of ESBL and Carbapenemase-Producing Bacteria

**DOI:** 10.3390/antibiotics9060286

**Published:** 2020-05-27

**Authors:** Ali F. Saleem, Ahreen Allana, Lauren Hale, Alondra Diaz, Raul Salinas, Cristina Salinas, Shahida M. Qureshi, Aneeta Hotwani, Najeeb Rahman, Asia Khan, Anita K. Zaidi, Patrick C. Seed, Mehreen Arshad

**Affiliations:** 1Department of Pediatrics and Child Health, Aga Khan University Hospital, National Stadium Rd, Karachi 74800, Pakistan; ali.saleem@aku.edu (A.F.S.); ahreen_allana@hotmail.com (A.A.); shahidamqureshi@gmail.com (S.M.Q.); Aneeta.hotwani@aku.edu (A.H.); najeeb.rahman@aku.edu (N.R.); asia.khan@aku.edu (A.K.); anita.zaidi@aku.edu (A.K.Z.); 2Department of Pediatrics, Duke University, 2301 Erwin Rd, Durham, NC 27710, USA; laurenehale1@gmail.com (L.H.); raul.salinas@duke.edu (R.S.); cristinasalinas95@gmail.com (C.S.);; 3Ann & Robert H. Lurie Children’s Hospital, 225 E Chicago Ave, Chicago, IL 60611, USA; alodiaz@luriechildrens.org (A.D.); pseed@luriechildrens.org (P.C.S.); 4Stanley Manne Children’s Research Institute, 303 E Superior St, Chicago, IL 60611, USA; 5Northwestern Feinberg School of Medicine, 420 E Superior St, Chicago, IL 60611, USA

**Keywords:** Enterobacteriaceae, multi-drug resistant, extended-spectrum beta-lactamase, carbapenemase, infants, gut colonization

## Abstract

The recent rapid rise of multi-drug resistant Enterobacteriaceae (MDR-E) is threatening the treatment of common infectious diseases. Infections with such strains lead to increased mortality and morbidity. Using a cross-sectional study, we aimed to estimate the prevalence of gut colonization with extended spectrum beta-lactamase (ESBL) producing Enterobacteriaceae among healthy infants born in Pakistan, a setting with high incidence of MDR-E infections. Stool samples were collected from 104 healthy infants between the ages of 5 and 7 months. Enterobacteriaceae isolates were screened for resistance against several antimicrobial classes. Presence of ESBL and carbapenemase genes was determined using multiplex PCR. Sequence types were assigned to individual strains by multi-locus sequence typing. Phylogenetic analysis of *Escherichia coli* was done using the triplex PCR method. Forty-three percent of the infants were positive for ESBL-producing Enterobacteriaceae, the majority of which were *E. coli*. We identified several different ESBL *E. coli* sequence types most of which belonged to the phylogenetic group B2 (23%) or D (73%). The widespread colonization of infants in a developing country with ESBL-producing Enterobacteriaceae is concerning. The multiple sequence types and reported non-human sources support that multiple non-epidemic MDR lineages are circulating in Pakistan with healthy infants as a common reservoir.

## 1. Introduction

Gram-negative rods (GNR) such as *Escherichia coli* are a major cause of neonatal infections [1,2]. Within all six WHO regions, individual countries have reported >50% resistance rates against third-generation cephalosporins among the Enterobacteriaceae *E. coli* and *K. pneumoniae*. This global rise in antimicrobial resistance among GNR is limiting our ability to treat such infections. The rapid spread of the highly resistant *E. coli* strain ST131 underscores the fact that such organisms are no longer confined by geographical boundaries [3]. In low middle-income countries (LMIC), neonates and infants, who are infected with multi-drug resistant Enterobacteriaceae (MDR-E), have higher mortality likely attributed to a delay in appropriate therapy [4]. In high-income countries (HIC), MDR-E infections significantly increase neonatal morbidity, length of hospital stays and health-care costs [5]. Both maternal and infant gut colonization with pathogenic *E. coli* strains may be associated with a higher risk of bloodstream infections in infants [6]. Therefore, colonization with an MDR-E strain could be associated with an additional risk of mortality and morbidity.

Pakistan is a lower middle-income country with the highest rate of infant mortality in the world (46/1000 live births). Infections are among the leading causes of infant mortality [7]. Antibiotic overuse is common and is attributed to frequent self-medication, inappropriate prescription practices and excessive antibiotic use in livestock [8,9,10]. As a result, antimicrobial resistance has rapidly increased.

Most infection control strategies used to mitigate the spread of MDR-E are focused on in-patient facilities or adults with comorbidities, antibiotic use or previous hospital exposure in the outpatient setting. Healthy infants in the community are usually not considered an important reservoir of MDR-E since they typically do not have the risk factors known to be associated with acquisition of MDR-E. However, recent studies have shown that certain MDR-E are adept in colonizing the healthy gut even in the absence of antibiotic exposure [11,12]. Using a retrospective collection of stool samples from healthy infants, we aimed to conduct a cross-sectional study to estimate the prevalence of gut colonization with ESBL-producing Enterobacteriaceae among infants in a community with a high incidence of MDR-E infections.

## 2. Methods

### 2.1. Patient Enrollment and Sample Collection

We analyzed the stool samples of 104 healthy infants between the ages of 5–7 months. The patient population was from a multicenter randomized control trial titled “Polio End-Game Strategies—Poliovirus Type 2 Challenge Study, a Five Arm Community-Based Randomized Controlled Trial, Karachi, Pakistan” [13]. This study was conducted at Primary Health Care (PHC) centers located in four different peri-urban communities of Karachi: (1) Bhains Colony, (2) Rehri goth, (3) Ali Akbar Shah Colony and (4) Ibrahim Hyderi. Stool specimens were collected at the PHC center or at the children’s homes and stored at +4 °C prior to same-day transport to the Pediatric Infectious Disease Research Lab (PIDRL) at Aga Khan University in Karachi, Pakistan. Demographic data regarding birth weight, total household income, parental educational level, household members, inter-current diarrhea and breastfeeding status were collected.

Infants were excluded if they were born prematurely (at <37 weeks gestation), had a confirmed bacterial or viral diarrheal illness in the last 4–6 weeks, were acutely ill, or had a history of prolonged hospitalization.

### 2.2. Bacterial Isolate Testing and Antimicrobial Sensitivities

Stool culture was done at PIDRL. To select GNR, stool samples were plated on MacConkey agar and incubated at 37 °C in room air for 24 h. The most common morphological colony-types were identified using standard biochemical methods including enzyme-based tests for catalase, oxidase, indole and urease [14]. Susceptibility against common antibiotics was determined using the disk diffusion method and by the Clinical Lab Standards Institute (CLSI) guidelines [15]. Screening for Extended Spectrum Beta-Lactamase (ESBL) resistance was performed by a disk diffusion method using Ceftazidime, Cefotaxime, Ceftriaxone and Clavulanate using CLSI cutoffs [15].

### 2.3. DNA Extraction and Molecular Testing for ESBL and Carbapenemases

Genomic DNA from bacterial isolates was extracted using Wizard Genomic DNA Purification Kit (Promega, Madison, WI, USA). Whole DNA from stool was extracted using the MO BIO PowerSoil DNA Isolation Kit (Qiagen, Germantown, MD, USA). Multiplex PCR was done using primers designed against conserved regions of the ESBLs and carbapenemases as previously described [16].

### 2.4. Multi-Locus Sequence Typing (MLST)

Sequence typing of the ESBL-producing *E. coli* isolates was done using the Pasteur method [17]. Briefly, standard primers were used to amplify 8 housekeeping genes in *E. coli*; *dinB* (DNA polymerase), *icdA* (isocitrate dehydrogenase), *pabB* (p-aminobenzoate synthase), *polB* (polymerase PolII), *putP* (proline permease), *trpA* (tryptophan synthase subunit A), *trpB* (tryptophan synthase subunit B), *uidA* (beta-glucuronidase). These primers have a universal 3’ tail used for sequencing. Amplicons were extracted, gel purified with GeneJet Gel Extraction Kit (Thermofisher Scientific, Waltham, MA, USA) and sequenced using Sanger sequencing at Genewiz, Inc (South Plainfield, NJ, USA). Sequence types were assigned by matching individual allele sequences and allele combination with those available in the Pasteur MLST database (http://bigsdb.pasteur.fr/ecoli/).

### 2.5. Phylogenetic Analysis

Phylogenetic analysis of *E. coli* was done using a triplex PCR method previously described by Clermont et al. [18]. Briefly, this method utilizes a combination of two genes (1) *chuA*, a gene involved in heme transport among enterohemorrhagic O157:H7 *E. coli*; (2) *yjaA*, a gene initially identified in *E. coli* K-12, and whose function is as yet unknown; and (3) an anonymous DNA fragment designated TSPE4.C2 from a subtractive library which aimed to identify conserved regions among *E. coli* strains involved in neonatal meningitis [19]. PCR conditions previously described were used along with 2 µL of extracted DNA from individual bacterial isolates.

### 2.6. Statistical Analysis

Statistical analysis was done using Stata 15.0 (Stata Inc., College Station, TX, USA). Median and interquartile range were calculated for quantitative variables like age and birth weight while frequency and percentage were calculated for qualitative variables like gender, history of breastfeeding and organism. Results were tabulated using Microsoft Office Excel 2016.

### 2.7. Ethical Approval

The study was approved by the Ethical Review Committee of Aga Khan University, the National Bioethics Committee of Pakistan, and the Institutional Review Board at Duke University. All activities followed the guidelines of good clinical practice; the original trial protocol was registered at ClinicalTrials.gov (identifier NCT02189811).

## 3. Results

We conducted a cross-sectional analysis of 104 stool samples previously collected for a Polio vaccine efficacy trial. The trial enrolled healthy infants from the community in a peri-urban area near the metropolitan city of Karachi, Pakistan. Infants were given routine polio vaccination soon after birth and the vaccine IgA response was evaluated over the next few months. No further intervention was done in this population, therefore stool samples from this cohort represented a readily available sample collection for our study. Table 1 shows the available demographic details of the enrolled subjects. Similar percentages of males and females (55% vs. 45%) mostly between 5 and 7 months of age participated. The majority (69%) of infants were exclusively breastfed. Among those that were receiving supplemental nutrition, the majority were being fed formula milk.

Figure 1A summarizes the distribution of ESBL and non-ESBL isolates among the three most commonly isolated Enterobacteriaceae. Forty-three percent (45/104) of the infant stool cultures yielded ESBL-producing Enterobacteriaceae (ESBL-E), 84% of which were *E. coli*, 10% were *Klebsiella* spp., and 6% were *Enterobacter* spp. Among the *E. coli* isolates approximately 80% (62/76) were ESBL producers.

Molecular testing of the ESBL-producing *E. coli* showed that 27 were positive for CTX-M ESBL gene alone and 7 were positive for TEM ESBL gene alone. Both TEM and CTX-M genes were detected in 26 of the isolates and 2 isolates had neither. None of the strains were positive for carbapenemase genes. We also noted that 37% of all infants positive for ESBL-E carried 2 or more ESBL-E strains.

Figure 1B,C show the antibiotic susceptibility patterns for ESBL and non-ESBL stool *E. coli* isolates respectively. World Health Organization (WHO) guidelines indicate ampicillin and third-generation cephalosporins as empiric treatment choices for neonatal sepsis along with aminoglycoside [20]. However, in the study population, ampicillin and third-generation cephalosporin resistance was ubiquitous among the ESBL strains. Among non-ESBL strains, ampicillin resistance was found in approximately 61% of the isolates while approximately 31% were resistant to amoxicillin-clavulanate. Resistance against aminoglycosides, carbapenems and macrolides was relatively low among both ESBL and non-ESBL strains.

While culture-based methods are more reliable for identifying a drug resistance profile for individual strains, it may not be sensitive enough to detect all resistant strains found in the infant stool samples. Consequently, molecular testing on whole stool DNA samples was also done to determine the presence of ESBL and carbapenemase genes. TEM was the most common (70%), followed by CTX-M (55%) and VEB (16%), as shown in Figure 2A. More than 25% of the infant stools were also positive for the carbapenemase gene KPC. Stool samples also tested positive for additional carbapenemase genes including DIM (16%), AIM (16%) and OXA (8%). Figure 2B illustrates the number of resistance genes harbored by individual infants. Molecular testing for resistant genes suggested that 80% (83/104) children were carriers of ESBL and/or carbapenemase genes, with 65% of infants carrying more than two resistance genes.

The sixty-two ESBL-producing *E. coli* isolated from the infant cohort were also sequence-typed. Using the Pasteur MLST method, 46 of the *E. coli* isolates were assigned to a known sequence type. Sixteen isolates could not be matched to a fully sequenced strain in the Pasteur MLST database. We hypothesized that these strains either represented unique polymorphisms of previously identified strains or previously unidentified strains that were specific to the South Asian region and not present in the MLST database. Table 2 lists the strains, their frequency of isolation, the method of identification, the phylogenetic group and any previously reported source. Except for ST8 which was found in 11 infants, most other strains have a more limited distribution in 1–5 infants, suggesting that the infants in this community cohort were colonized with diverse ESBL-producing *E. coli* strains despite the modest overall sample size and relatively constrained geographic distribution of the infant cohort. Appendix A shows the geographic distribution of the ESBL isolates.

Phylogenetic grouping of the *E*. *coli* strains determined that other than ST681 that belonged to group A, all other sequence types isolated from infants in our cohort belonged to either group B2 (6/26) or group D (19/26). We also searched the Enterobase database for the previously reported source of each of the 26 sequence types identified. As shown in Table 2, the majority of these strains have been previously isolated from humans, animals and the environment, suggesting that these strains were adapted to survive in a variety of hosts.

## 4. Discussion

Our study demonstrates prevalent colonization of healthy, community-based Pakistani infants with ESBL and carbapenemase-producing Gram-negative bacteria. Previous studies from diverse regions of the world have reported healthy infants as carriers of multi-drug resistant bacteria, though only two recent reports from Lebanon and Bangladesh have a similar prevalence of ESBL genes as found in our infant cohort [21,22]. Unlike any previous study, molecular analysis of fecal samples from our Pakistani cohort showed that approximately 25% of infants harbored carbapenemase-encoding bacteria.

Monira et al. reported colonization with MDR Gram-negative bacteria in a cohort of 15 healthy infants, 10–24 months of age from Bangladesh [22]. They acquired a total of 122 bacterial isolates from all infant stools under aerobic conditions, of which 73% were found to be multi-drug resistant. Similar to our findings, *E. coli* was the predominant MDR Gram-negative. Among all bacterial isolates, the most common beta-lactamases were TEM (26%) and CTX-M1 (32%). Hijazi et al. reported that among 117 healthy Lebanese infants between 1–12 months of age, 49% were colonized with ESBL-producing Enterobacteriaceae [21]. CTX-M type beta-lactamases were the most common followed by TEM beta-lactamase. Male gender, hospital birth, cesarean delivery and being formula-fed were associated with a greater likelihood of being colonized with ESBL-producing Enterobacteriaceae.

Colonization with MDR Gram-negative bacteria precedes most invasive infections including pneumonia, bloodstream and urinary tract infections [23]. Data from Pakistan show extremely high rates of community-acquired infections with Enterobacteriaceae that are resistant to third-generation cephalosporins as well as carbapenems [24]. More concerningly, there has been a rapid increase in infections with carbapenem-resistant Gram-negative bacteria [25]. One report showed an almost 16% resistance against Colistin among carbapenem-resistant Enterobacteriaceae (CRE) isolates processed by a tertiary-care hospital associated clinical microbiology lab [26]. Currently, there is an outbreak of extensively drug-resistant *Salmonella typhi* shown to harbor an IncY plasmid that encodes beta-lactam and fluoroquinolone resistance genes [24].

*E. coli* was the most common Gram-negative Enterobacteriaceae isolated from the infant stools. Within this age group, *E. coli* plays a significant role in invasive infections, including sepsis, meningitis and urinary tract infections [2]. Recent data suggest that gut colonization with Gram-negative bacilli can increase the risk of subsequent invasive infections in infants [6]. Therefore, colonization with *E. coli* resistant to beta-lactams, which are recommended as first-line therapy by WHO [27], may increase the risk and prevalence of therapeutic failure or delayed efficacious therapy.

ESBL enzymes such as TEM, SHV, CTX-M, OXA, and VEB are part of group 2 serine beta-lactamases [28]. SHV and, the widely spread, TEM enzyme was among the first to be recognized from this group. However, CTX-M enzymes are now rapidly becoming a common mechanism of bacterial resistance in many parts of the world including South Asia [29]. The increasing trend in CTX-M type ESBLs is one of large concern as these pose a threat to the use of third-generation cephalosporins. In fact, up to one-fifth of healthy individuals living in South Asia or with a travel history to South Asia are carriers of CTX-M-producing bacteria [30]. These rates are considerably lower than the prevalence of CTX-M genes in our infant stool cohort and such a high prevalence of TEM or CTX-M genes have not been reported previously from the South Asian region among any age group.

The OXA-type beta-lactamases are most commonly found in North Africa, South America, parts of Europe and the Middle East and South-East Asia [31]. The infrequently identified VEB beta-lactamase is primarily limited to South-East Asia, in particular Vietnam where it was first discovered, and neighboring Thailand [32,33]. Recently, however, VEB-producing GNRs including *E. coli* and *P. aeruginosa* were detected in China, India and Bangladesh [34,35]. There are no reports of such strains being detected in Pakistan but our data suggests that these strains may indeed be present within the Pakistani community as well.

Gut colonization with carbapenemase-producing bacteria has not been previously reported among infants and children without prior hospital contact. The carbapenemase genes tested for in our study belong to both group 2 serine beta-lactamases, the most common of which is the KPC gene, and group 3 metallo beta-lactamases, including NDM, VIM, IMP and DIM [28]. While infants in our cohort did not have a significant prevalence of NDM genes (4%), it was interesting to note a relatively high prevalence of DIM carbapenemase gene, an uncommon metallo beta-lactamase. This enzyme has previously been associated with carbapenem-resistant *Pseudomonas* but has not been identified in healthy hosts [36]. About one-fourth of the infant stools in our cohort were positive for the KPC gene. This finding is highly concerning because infants who are positive for the KPC gene are also likely colonized with carbapenem-resistant bacteria. Further, the KPC gene is often part of the Tn3-type transposon, *Tn4401*, which is capable of inserting itself into diverse plasmids carried by Gram-negatives, and is frequently found on plasmids that confer resistance to multiple classes of antibiotics [37].

We identified isolates with a wide variety of sequence types suggesting the limited spread of epidemic strains in this region. Since infants included in our study were from four separate villages around the major urban city of Karachi, we do not suspect local outbreaks of these strains. It was interesting to note that many of these strains had previously been isolated from diverse hosts, such as humans, livestock and poultry, as well as from the environment. Data regarding animal exposure were not collected in the original study, however, according to a survey done by Food and Agriculture Organization of the United Nations, many families living in the peri-urban and rural villages in Pakistan keep a small number of chickens, goats or cows as a personal food source [38]. We do not expect most infants at approximately six months of age to come into direct and frequent contact with household animals; though such animal-human transmission events are likely happening in the adults in the families. Infants can then acquire antibiotic-resistant strains either perinatally, through breast milk or postnatally through environmental exposures [39].

Most strains isolated from our infant cohort belonged to either phylogenetic group B2 or D. Most extra-intestinal pathogenic *E. coli* (ExPEC) belong to group B2 and some to group D [40]. A study in Swedish infants showed that group B2 *E. coli* can persist in infant microbiome independent of associated virulence factors and may be more adept in prolonged colonization than strains belonging to other phylogenetic groups [41]. Some ExPEC strains, such as the pandemic ST131, have been shown to out-compete commensal gut flora in a mouse model of gut colonization. However, the same has not been studied for other ExPEC sequence types [11].

There are several limitations to this study. We utilized a previously collected cohort of infant stool samples, thus limited data are available on maternal factors, such as infections during pregnancy, delivery method, infant anthropometric measurements; and on environmental exposures, such as household animals and water sources. Maternal and infant antibiotic use data were also not available. Furthermore, detailed data on prior hospitalizations were not available for the infants included in this study or their parents and close relatives. However, it is important to note that infants were excluded from this study if they had confirmed bacterial or viral diarrheal illness in the 4–6 weeks prior to sample collection, were acutely ill, or had a history of prolonged hospitalization. Taken together, a direct relationship between antibiotic exposure and colonization with antibiotic-resistant strains cannot be made. Since this was a cross-sectional study, only single samples were tested. A negative stool sample may, therefore, be due to limited sample amount collected and as a result we may underestimate the overall rates of MDR GNR colonization.

In summary, this study shows that healthy infants living in a community with a high incidence of antimicrobial resistance are a substantial reservoir of multi-drug resistant Enterobacteriaceae. These data suggest an urgent need to better understand the microbial, host and environmental determinants of gut colonization with MDR GNR especially in young infants in developing countries and improve strategies for infection control both in the hospital and the community. The spread of bacteria harboring these plasmids within the general population is a major public health concern. Ongoing studies will determine the persistence of these MDR organisms in the developing child and the relationship of MDR bacterial colonization with invasive infection and treatment failure.

## Figures and Tables

**Figure 1 antibiotics-09-00286-f001:**
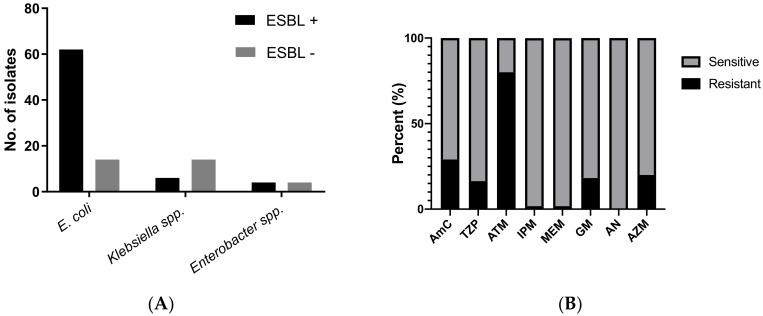
(**A**) Distribution of Extended Spectrum Beta-lactamase (ESBL) + and ESBL—isolates among the three most common Enterobacteriaceae. (**B**) Susceptibility pattern of ESBL *E. coli* isolates. Resistance against beta-lactams was ubiquitous, sensitivities against other common antibiotics was shown. (**C**) Susceptibility pattern of non-ESBL *E. coli* isolates against commonly used antibiotics. AM ampicillin, CRO ceftriaxone, CTX cefotaxime, CAZ ceftazidime, AmC amoxicillin-clavulanate, TZP pipercillin-tazobactam, ATM aztreonam, IPM imipenem, MEM meropenem, GM gentamicin, AN amikacin, AZM azithromycin.

**Figure 2 antibiotics-09-00286-f002:**
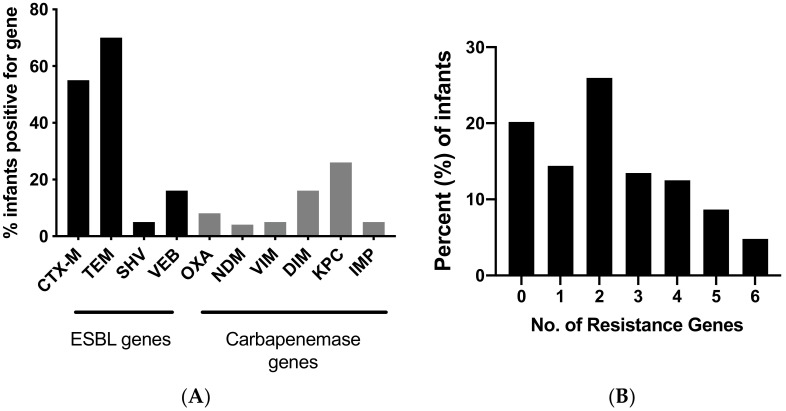
(**A**) Resistance genes carried by infants as determined by PCR testing for ESBL and carbapenemase genes. (**B**) Number of resistance genes per infant stool.

**Table 1 antibiotics-09-00286-t001:** Demographic details of the study population.

Demographic Parameter	N (%)
**Gender of child**	
Male	58 (55)
Female	46 (45)
n	104
**Age groups (months)**	
Median age	5.1
Interquartile range	0.36 (5.059–5.420)
**Birth weight (kg)**	
Median birth weight	2.83
Interquartile range	0.55 (2.56–3.11)
**History of breastfeeding**	
Exclusively breast fed	74 (69.2)
**Type of milk (if not exclusively breastfed)**	
Formula milk	21 (63.6)
Cow milk	5 (15.2)
Other (goat milk or weaning)	7 (21.2)

**Table 2 antibiotics-09-00286-t002:** Sequence type, identification, phylogenetic group and previously reported sources of ESBL+ *E. coli.*

Sequence Type	Frequency (No. of Isolates)	Identification Method ^1^	Phylogenetic Group	Reported Source ^2^
2	5	PUBMLST	B2	human/domesticated animal
4	1	PUBMLST	D	human/food
5	1	PUBMLST	D	human
8	11	PUBMLST	D	domesticated animal
21	1	PUBMLST	D	humans/livestock
87	1	PUBMLST	D	primate
132	2	PUBMLST	B2	animal
244	1	PUBMLST	D	human
316	2	PUBMLST	D	human/fish/poultry/livestock/environment
357	1	PUBMLST	D	human/poultry
398	1	PUBMLST	D	human/livestock/environment
399	1	PUBMLST	D	human/fish/poultry/livestock/environment
511	2	PUBMLST	D	human
535	1	PUBMLST	D	human
537	1	PUBMLST	D	human/livestock
551	1	PUBMLST	D	human
569	2	PUBMLST	B2	human/environment/poultry
573	1	PUBMLST	D	human
638	2	PUBMLST	B2	human
647	1	PUBMLST	D	human/livestock
661	2	PUBMLST	D	human/environment/livestock
681	1	PUBMLST	A	human
740	1	PUBMLST	D	human
756	1	PUBMLST	B2	animal
767	1	PUBMLST	B2	human/livestock
809	1	PUBMLST	D	human

^1^ Sequence type was identified using the Pasteur method for multi-locus sequence typing (PUBMLST). ^2^ Source noted in this column is as reported by investigators who had submitted the corresponding STs to Enterobase.

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
