# Peer review of "The Gut of Healthy Infants in the Community as a Reservoir of ESBL and Carbapenemase-Producing Bacteria"

_antibiotics, 2020, doi:10.3390/antibiotics9060286_

Round 1

Reviewer 1 Report

In this manuscript, the authors reported the antibiotic resistance profile of bacteria especially of ESBL isolated from fecal samples of infants in Pakistan. All the experiments have been conducted logically and the experimental design is well-suited. Overall, the manuscript is written well. Just a few minor corrections are required

Page 2 line 83 - complete address location of Promega

Page 2 line 84 - complete address location of Qiagen

Page 2 line 92 - complete address location of GeneJET

Page 3 line 123 - I believe it should be fig. 1B instead of 1A

Page 4 line 127 - I believe it should be fig. 1A instead of 1B

Page 4 fig. 1D - Based on author's description, it would be appropriate to change the y-axis to percentage.

Page 5 fig. 2B - Based on author's description, it would be appropriate to change the y-axis to percentage.

Author Response

We thank the reviewer for their comments and suggestions. Please see our responses below. 

Page 2 line 83 - complete address location of Promega. We have added the complete address. 

Page 2 line 84 - complete address location of Qiagen. We have added the complete address. 

Page 2 line 92 - complete address location of GeneJET. We have added the complete address. 

Page 3 line 123 - I believe it should be fig. 1B instead of 1A. This was an error on our part and has been rectified. 

Page 4 line 127 - I believe it should be fig. 1A instead of 1B. This was an error on our part and has been rectified. Please note that based on the second reviewers comments we have separated Figure 1B in to ESBL and non-ESBL E. coli isolates. 

Page 4 fig. 1D - Based on author's description, it would be appropriate to change the y-axis to percentage. Based on comments by reviewer 2 we have removed this figure and mentioned the results in the text of the manuscript. 

Page 5 fig. 2B - Based on author's description, it would be appropriate to change the y-axis to percentage. Thank you for this suggestions, we have changed the y-axis to percentage. 

Reviewer 2 Report

The authors assessed the prevalence of antibiotic resistant strains of Enterobacteriaceae in community environment. The methodology is clear, though there is place for some improvements.

I suggest to take account of previous hospitalizations of the children and their parents/close family members. If this is possible, present the data in Table 1 as demographic parameter. This could be important in order to connect the identified MDR strains with a possible hospital-related source. If not possible, please add as study limitations aside the other ones.

Figure 1A and B are switched considering the in-text references. Figures 1C and 1D are not required, and the related text could be improved for an easy following of the information.

I would split Figure 1A in 2 parts: antibiotic susceptibility pattern for non-ESBL and for ESBL (here do not include the beta-lactamins, as 100% resistance is present considering ESBL definition).

Figure 2A - please color the OXA bar according to Carbapenemase (or is it ESBL OXA? - not clear, because these are referred to in the discussions section).

The references look inappropriate to the journal's requirements.

Author Response

We thank the reviewer for their comments and suggestions. Please see are responses below: 

I suggest to take account of previous hospitalizations of the children and their parents/close family members. If this is possible, present the data in Table 1 as demographic parameter. This could be important in order to connect the identified MDR strains with a possible hospital-related source. If not possible, please add as study limitations aside the other ones. 

We agree with the reviewer that this data would have been very important, however unfortunately this was not collected at the time of initial enrollment of the infants. We have added this as a limitation to this study.

Figure 1A and B are switched considering the in-text references. Figures 1C and 1D are not required, and the related text could be improved for an easy following of the information.

Figure 1A and 1B were an oversight on our part so we thank the reviewer for identifying this error. We have further removed the previous 1C and 1D figures and added more details in the related text. 

I would split Figure 1A in 2 parts: antibiotic susceptibility pattern for non-ESBL and for ESBL (here do not include the beta-lactamins, as 100% resistance is present considering ESBL definition).

Figure 1A (which should have been figure 1B) has been separated in to two as suggested. 

Figure 2A - please color the OXA bar according to Carbapenemase (or is it ESBL OXA? - not clear, because these are referred to in the discussions section).

The OXA gene has been re-labeled as a carbapenemase. 

The references look inappropriate to the journal's requirements.

The references have also been corrected.